# Research Advances of d-allulose: An Overview of Physiological Functions, Enzymatic Biotransformation Technologies, and Production Processes

**DOI:** 10.3390/foods10092186

**Published:** 2021-09-15

**Authors:** Yu Xia, Qianqian Cheng, Wanmeng Mu, Xiuyu Hu, Zhen Sun, Yangyu Qiu, Ximing Liu, Zhouping Wang

**Affiliations:** 1State Key Laboratory of Food Science and Technology, Jiangnan University, Wuxi 214122, China; wmmu@jiangnan.edu.cn (W.M.); wangzp@jiangnan.edu.cn (Z.W.); 2School of Food Science and Technology, Jiangnan University, Wuxi 214122, China; 6200113010@stu.jiangnan.edu.cn (Q.C.); zhsun@jiangnan.edu.cn (Z.S.); 6200112152@stu.jiangnan.edu.cn (Y.Q.); liuximing0109@163.com (X.L.); 3Collaborative Innovation Center of Food Safety and Quality Control in Jiangsu Province, Jiangnan University, Wuxi 214122, China; 4China Biotech Fermentation Industry Association, Beijing 100833, China; hxy@cfia.org.cn

**Keywords:** d-allulose, sweetener, physiological functions, ketose 3-epimerase, directed evolution, enzymatic biotransformation, production processes

## Abstract

d-allulose has a significant application value as a sugar substitute, not only as a food ingredient and dietary supplement, but also with various physiological functions, such as improving insulin resistance, anti-obesity, and regulating glucolipid metabolism. Over the decades, the physiological functions of d-allulose and the corresponding mechanisms have been studied deeply, and this product has been applied to various foods to enhance food quality and prolong shelf life. In recent years, biotransformation technologies for the production of d-allulose using enzymatic approaches have gained more attention. However, there are few comprehensive reviews on this topic. This review focuses on the recent research advances of d-allulose, including (1) the physiological functions of d-allulose; (2) the major enzyme families used for the biotransformation of d-allulose and their microbial origins; (3) phylogenetic and structural characterization of d-allulose 3-epimerases, and the directed evolution methods for the enzymes; (4) heterologous expression of d-allulose ketose 3-epimerases and biotransformation techniques for d-allulose; and (5) production processes for biotransformation of d-allulose based on the characterized enzymes. Furthermore, the future trends on biosynthesis and applications of d-allulose in food and health industries are discussed and evaluated in this review.

## 1. Introduction

The prevalence of chronic diseases, such as diabetes, hypertension, and obesity, has increased rapidly across the globe in recent years. These diseases are partially caused by excessive dietary energy intake (e.g., high glycemic and fat dietary preferences), and pose serious threats to human health [1,2]. Seeking and applying low-calorie food supplements and functional food additives may be an attractive strategy to solve these problems. Therefore, low-calorie and functional rare sugars have attracted more and more attention in the research and development of foods, cosmetics, and medicines [3]. Among the rare sugars, d-allulose (also known as d-psicose (d-ribo 2-ketohexulose)) is considered to be a good alternative to sucrose, with a relative sweetness of 70% and very few calories [4]. It is a white odorless powdery crystal compound that crystallizes into β-d-pyranose via 1C (^1^C4 (d)) conformation [5]. The detailed chemical properties of d-allulose as a sweetener are illustrated in Table 1. d-Allulose is naturally found in wheat and processed sugar cane, and it is also found in mixtures of fructose and glucose syrups in trace amounts [6,7,8]. Experiments on the long-term effects of d-allulose ingestion have indicated no harmful effects in dogs [9]. It has been shown that d-allulose is partly absorbed in the small intestine and is released into the bloodstream [3], while a small portion of d-allulose is delivered to the large intestine, and has been found to be partially fermented in the appendix in rats and, to a lesser extent, in human intestines [4]. Studies have demonstrated that d-allulose has many positive physiological effects, for instance, regulating lipid metabolism [10,11], neuroprotection [12], and anti-obesity [13,14], indicating great potential for applications in foods and food ingredients. Furthermore, it can enhance the properties of gel strength, emulsion stability, foaminess, and oxidation resistance in egg-white protein containing food products through the modulation of Maillard reactions [15].

The synthesis of d-allulose through chemical or biological methods has been a hot topic in recent years. There are mainly two chemical production processes for this compound: one is synthesis from glucose or 1,2:4,5-di-O-isopropylidene-β-d-fructopyranose under high-temperature reaction conditions catalyzed by molybdate [16,17], and the other is the preparation of d-allulose from fructose by heating in a mixed system of alcohol and triethylamine, with multi-step reactions [18]. Because of the shortcomings of complex purification procedures, the generation of numerous toxic by-products, and their further environmental pollution, these two processes are not suitable for the industrial production of d-allulose. Unlike chemical synthesis processes, enzymatic biotransformation approaches have the advantages of simplicity, green environmental effects, and lower costs. These processes are mainly based on C-3 epimerization reactions, which transform the substrate d-fructose to the product d-allulose by ketose 3-epimerase [19,20,21].

Several ketose 3-epimerases have been isolated and identified from different microbial origins. Most of them are weakly basophilic [22,23], except for *Dorea* sp. CAG317 d-psicose 3-epimerase (DPEase) and *Christensenella minuta* DPEase, which posess a weakly acidic optimal pH [24,25]. Ketose 3-epimerases have been heterologously expressed successfully in prokaryotic and eukaryotic systems for the efficient production of D-allulose and its high-value derivates. The Gibbs free energy change ΔG value is approximately 0.1 kJ/mol for the transformation of d-allulose from d-fructose (http://equilibrator.weizmann.ac.il/ accessed on 13 August 2021) [26], suggesting that, theoretically, around half of d-fructose can be converted to d-allulose, whereas the actual conversion ratio is obviously below the theoretical value [27]. One of the reasons for the low conversion ratio is the inferior thermal stability of these enzymes. Recently, sophisticated techniques and approaches have been utilized to increase the yield of d-allulose. For example, positive results were obtained by screening different enzymes with a superior heat tolerance through gene mining and directed evolution or using borate to break the equilibrium of the reaction [28], as well as the separation of d-allulose from reaction solutions by simulated moving bed technology (SMB) [29,30]. Over the past two decades, with the maturation of molecular and genetic engineering techniques, such as interfacial property analysis and B-factor iterative testing [31], the mutagenesis and property improvements of d-allulose 3-epimerase have been successfully fulfilled. It is also an efficient strategy to extend the half-life of the enzymes and to maintain their stability through immobilization of the ketose epimerases on novel materials (iron oxide magnetic nanoparticles, graphene oxide, etc.) [32,33]. Meanwhile, studies on the enzymatic production of rare sugars derived from cheap raw materials (such as starch, fruit juice, and inulin) are ongoing, aiming to establish a new strategy for the industrial production of d-allulose with a low energy consumption and economic efficiency.

In this review, we focus on the recent research advances of d-allulose, i.e., its physiological functions, enzymatic biotransformation technologies, and production processes, in order to provide an overview of the current research progress.

## 2. Physiological Functions of d-allulose

d-Allulose has been approved by the US Food and Drug Administration (FDA) as the “generally recognized as safe” status (GRAS; GRN nos. 400, 498, 624, 647, 693, and 755) and has been permitted as an ingredient in various foods and dietary supplements. When d-allulose is ingested by human bodies, 70% of it is absorbed in the small intestine with some fermentation from the intestinal bacteria, while 30% of it is excreted in the feces [34]. The absorbed d-allulose is not metabolized in the human body, but can be found in the urine [35]. A study that administered a d-allulose intake of 5 g per meal, for three meals a day for 12 weeks to healthy subjects with normal blood glucose levels found no liver damage or physical symptoms [36]. The LD_50_ value of d-allulose in rats was assessed to be 16 g/kg, while the maximal atoxic concentration in the human gastrointestinal environment was estimated to be 0.55 g/kg [37]. An et al. confirmed that d-allulose from a microbial origin had no significant toxicity to mice in a 90-day oral toxicity assay, although some adverse effects were found [38].

d-Allulose has also been demonstrated to have several beneficial effects on human health. It has been reported to have anti-obesity [39], postprandial blood glucose levelreducing [40], anti-diabetes [41], and anti-atherosclerosis [42] functions. A schematic diagram of the multiple physiological functions of d-allulose is illustrated in Figure 1. For example, d-allulose inhibits the expression of monocyte chemotactic protein-1, showing therapeutic effects against atherosclerosis [43]. It also interacts closely with gut microbes to reduce inflammatory symptoms, perhaps partially by decreasing the expression levels of gene Gm12250 in several tissues, and by increasing the microbial genera *Lactobacillus* and *Coprococcus* in the gut microbiota composition [44].

### 2.1. Anti-Obesity

In vivo, d-allulose can inhibit the synthase activity of fatty acid production or increase energy expenditure to reduce abdominal fat accumulation in rats, and can enhance postprandial fat oxidation to alleviate body weight in healthy humans [45]. Rats fed with a diet containing 3% d-allulose for 18 months showed significantly lower weight gain and a reduction of abdominal adipose tissue weight without side effects [46], compared with those controls fed with the same levels in a sucrose diet. One possible mechanism for the involvement of d-allulose in lipid metabolism is to regulate the adipogenic transcription factors for the inhibition of 3T3-L1 preadipocyte differentiation and lipid accumulation. With the presence of d-allulose in the diet, the expression levels of fat formation-related proteins are markedly decreased, including fatty acid synthases and adipocyte fatty acid-binding protein, while the intracellular triglyceride content is also visibly reduced with d-allulose treatment [47].

Furthermore, it is very effective at preventing high-fat diet (HFD)-induced obesity, hepatic steatosis, and dyslipidemia, as well as at significantly reducing the amount of visceral fat in HFD-induced obese rats [48]. d-Allulose can normalize the metabolic status of diet-induced obesity by altering the activity of lipid-regulating enzymes and their gene expression levels [45]. It has been demonstrated that dietary d-allulose modulates cholesterol metabolism in hamsters, partly by decreasing the serum proprotein convertase subtilisin/kexin type 9 (PCSK9) levels. Therefore, it is expected that d-allulose intake is beneficial for the improvement of cholesterol metabolism, leading to a reduced risk of atherosclerosis [49]. Choi et al. compared the effects of d-allulose with two kinds of probiotics in diet-induced obesity (DIO) mice and found that a synbiotic mixture containing d-allulose was more active at suppressing DIO [50]. Itoh et al. assessed the anti-obesity effects in d =-allulose-treated Lep (ob)/Lep (ob) mice without exercise therapy or dietary restriction, and the results suggested that dietary supplementation with d-allulose specifically affected postprandial hyperglycemia and obesity-related hepatic steatosis [51].

It has also been shown that d-allulose-regulated metabolites are involved in metabolic pathways for the β-oxidation of fatty acid and cholesterol to bile acid conversion regulations. Moreover, the possible d-allulose-altered glucuronide/xylose pathways have been identified [10].

### 2.2. Anti-Diabetes

Prolonged administration of d-allulose-containing syrup has suggested that it can induce the translocation of hepatic glucokinase from the nucleus to the cytoplasm, as it maintains glucose tolerance and insulin sensitivity in rats so that the blood glucose levels are maintained [41,52]. Such effects have reduced metabolic disturbances and cognitive impairments in male Wistar rats [10]. d-Allulose inhibits intestinal α-glucosidase and suppresses the glycemic response upon carbohydrate ingestion in rats [53]. As a substrate for glucose transporter protein 5 (GLUT5) in the small intestine, d-allulose enters and leaves the small intestinal cells via the glucose transporter GLUT5 and GLUT2, respectively [42], and its long-term effects are concentrated on preventing the production and development of T2DM in diabetic rats [54]. It has been found to reduce postprandial glucose levels in healthy subjects, but also in those subjects with borderline diabetic symptoms. Glucagon-like peptide 1 (GLP-1) secretion in rats is potently stimulated by d-allulose, and the potent effect of d-allulose on GLP-1 secretion has been found with injection into the intestinal lumen rather than the peritoneum, indicating that the sugar exerts its effect in the lumen of the small intestine [55]. A probable mechanism for the induction of GLP-1 by d-allulose has been found through activation of vagal afferent signaling, which reduces food intake and promotes glucose tolerance in model animals [56].

Researchers have found that d-allulose also prevents the onset and progression of T2DM for 60 weeks by maintaining blood glucose levels, reducing body weight growth rate, and controlling postprandial hyperglycemia while reducing HbA1c levels, compared with untreated control rats [57]. This improvement in blood glucose control was accompanied by maintenance of the plasma insulin levels and preservation of pancreatic beta cells, as well as a significant reduction in inflammation levels. Hossian et al. investigated the effects of d-allulose on insulin resistance in type 2 diabetic rats, and the results suggest that it protected B islets in the pancreas to improve insulin resistance [41]. d-allulose enhances hepatic HDL-cholesterol uptake via SR-B1 in primary rat hepatocytes [58]. Thus, it is capable of regulating lipid metabolism and reducing the risk of atherosclerosis. In addition, d-allulose can inhibit the activities of intestinal sucrase and maltase, which potentially reduce postprandial hyperglycemia.

### 2.3. Anti-Oxidation

Compared with other rare sugars, d-allulose is more effective at scavenging reactive oxygen radicals and has a better scavenging activity for reactive oxygen species (ROS) than d-glucose and d-fructose [59]. d-allulose-treated rats have revealed that it prevents bis (2-Ethylhexyl) phthalic acid-induced testicular damage by inhibiting ROS production [60]. Furthermore, it is neuroprotective against 6-hydroxydopamine-induced apoptosis on PC12 cells, and may play a potential neuroprotective role in the treatment of neurodegenerative diseases by increasing intracellular glutathione levels [12]. Soy protein isolates that are glycated with d-allulose can considerably improve functional properties such as solubility, antioxidant functions, and emulsification activities [61]. The possible mechanisms of various physiological functions of d-allulose are summarized in Table 2.

### 2.4. Application of d-allulose in Food Industries

Due to its neutral sweetness, reductive properties, and high browning reactivity, d-allulose is suitable not only for baked goods, but also for sauces, ketchup, confectionery, beverages, and other products [13]. The addition of d-allulose can make products produce a stronger water holding capacity in foods compared with that of sucrose. Soy gels with added d-allulose have a remarkable impact on digestive behavior, a property that can be used to design low-calorie, satiety enhancing confectionery products [62,63]. Unlike sucrose and sorbitol, heating of myofibrillar protein with d-allulose facilitates the formation of both disulfide and non-disulfide crosslinks, which may be related to the mechanical properties and water holding capacity of d-allulose gels [64]. Functional foods and formulas for special medical purposes that utilize d-allulose in the Maillard reaction with proteins are effective in the prevention of dental caries and related diseases caused by oxidative stress. A recent longitudinal study on d-allulose, in which the lifespan of *Caenorhabditis elegans* was increased under both monogenic and axenic culture conditions, found increased resistance to oxidative stress by d-allulose through a dietary restriction mechanism [65]. These results suggest that d-allulose is an excellent candidate for dietary restriction mimetics.

Interestingly, research on d-allulose has also indicated that the growth of several types of lactic acid bacteria in milk are influenced by this additive, suggesting that this rare sugar can be applied for regulating acid production in over-fermented milk [66]. These results suggest that d-allulose may play an important role in the production of dairy products. It also has significant application potentials in the sugar industry, as it is a precursor to d-allose [67], d-allitol [68], and other precious sugars or sugar alcohols.

## 3. Enzymatic Biotransformation of d-allulose

Currently, the Izumoring strategy provides researchers a novel approach to the enzymatic synthesis of d-allulose, and this is the dominant methodology for the production of rare sugars [69]. Particularly, ketose 3-epimerase has become a predominant catalyst for the biotransformation of d-allulose, which has a unique catalytic mechanism that is significantly different from other epimerases [70]. The ketose 3-epimerase family enzymes modify the substrates with a C3-O3 proton exchange mechanism rather than phosphorylation [71], which was confirmed by sufficient evidence through resolving the complex crystal structure of ketose 3-epimerase bounded with substrate and co-substance, respectively [72]. The detailed process for the enzymatic biotransformation of d-allulose is depicted in Figure 2, from gene mining to mutational modification, with a summary of the methods based on ketose 3-epimerases for producing d-allulose, as well as other rare sugars. Alternatively, as a complementary pathway to the Izumoring strategy, hydroxylation addition reactions catalyzed by aldolases relying on dihydroxyacetone phosphate (DHAP) [73], or strategies relying on dephosphorylation and phosphorylation cascades reactions, are also considered useful approaches for the synthesis of d-allulose [74].

### 3.1. Source of Enzymes

As the major catalyst for the biotransformation production of d-allulose, ketose 3-epimerases catalyze the interconversion of both d-fructose to d-allulose and d-tagatose into d-sorbose, through the mechanisms shown in Figure 3.

Isomerases of microbial origins have distinct substrate specificities and are mainly classified as the ketose 3-isomerases family as d-tagatose 3-epimerase (DTEase), d-psicose 3-epimerase (DPEase), or the later described d-allulose 3-epimerase (DAEase), according to the principle of their optimal substrate. Ketose 3-epimerases from diverse microbial sources, including *Pseudomonas cichorii* [75], *Rhodobacter sphaeroides* [76], and *Clostridium scindens* [77], have been isolated and identified from 1993 to the present. For example, Li et al. reconstituted a ketose 3-epimerase from *Caballeronia fortuita* and characterized it as a DTEase. The recombinant enzyme showed the highest activity at pH 7.5 and 65°C in the presence of Co^2+^, with a superior thermostability and broad substrate specificity, for which enzyme exhibited the highest substrate specificity towards d-tagatose compared with other reported ketose 3-epimerases [78].

Meanwhile, Kim et al. [19] isolated an enzyme of the DTEase family from *Agrobacterium tumefaciens* and named it DPEase, for its strong substrate specificity for d-allulose. Since then, several kinds of DPEases, such as those from *C. cellulolyticum* [79], *Ruminococcus* sp. [80], *Desmospora* sp. [81], *Bacillus* sp. [82], and *Rhodopirellula baltica* [31], have been screened and investigated. Patel et al. identified a novel d-allulose 3-epimerase gene (*dae*M) from a macroeconomics resource in a hot water reservoir. It showed an extremely elevated thermostability at 60 °C to 70 °C, which was the highest thermal stable d-allulose 3-epimerase that had been characterized yet [83]. The *dae*M gene was expressed intracellularly in *Bacillus subtilis* and the biotransformation reaction was performed with the whole-cells at 60 °C, which produced approximately 196.0 g/L d-allulose from 700.0 g/L d-fructose.

The detailed microbial sources and the related information for DTEases and DAEases are given in Table 3. The isomerization capacity of different ketose 3-epimerase varies considerably depending on its microbiological origin and reaction conditions (i.e., reaction temperatures, optimum pH, substrates, and metal ion concentrations). Notably, only the DPEase from *Dorea* sp. CAG317 and the DPEase from *C. minuta* are optimally adapted to weak acidic pH conditions [24,25], suggesting that these two enzymes have potential for application in the large-scale production of d-allulose.

### 3.2. Structural Features of the Enzymes

Currently, several ketose 3-epimerases crystal structures are available in the Protein Data Bank (PDB), which are derived from *A. tumefaciens* [96], *P. cichorii* [71], *C. cellulolyticum* H10 [72], and *Staphylococcus aureus* [97], and have been successfully analyzed. Despite the low phylogenetic relationships between the DAEase enzymes (Figure 4), the highly conserved active center metal ion binding sites and the key residues in substrate binding sites lead to the similar enzymatic properties of these enzymes. The structure-based multiple sequence comparison of DTEase/DAEase from different microorganisms is shown in Figure 5A. As for the enzyme structures, *A. tumefaciens* DPEase (PDB:2HK0) and *C. cellulolyticum* DTEase (PDB:3VNI) are both tetramers consisting of four identical subunits arranged in an asymmetric formation. Each of these four dimers consists of two subunits that interact in a tightly linked manner. In the tetramer, two dimers are exposed at the ends of the TIM barrel, providing closure at the edge of this TIM barrel. The active site hydrophobic grooves are located between two adjacent subunits, which bind the substrate for the front end of the dimer to form a good contact surface. Its active center binding site is an octahedral ligand consisting of two water molecules and four amino acid binding sites (Glu, Asp, His, and Glu), while these residues are strictly conserved in almost all ketose 3-epimerases [96]. The substrate replaces the water molecule in the enzyme’s active center to bind to the enzyme and form an enediol cis-enediolate intermediate, while His188 and Arg217 enhance the stability of the intermediate when the enzyme catalyzes the reaction [71]. This indicates that metal ions play a critical role in the binding of substrate and enzyme during enzyme-catalyzed reactions, hence the majority of d-allulose 3-epimerases have a divalent metal cation dependence. The crystal structure of the engineered d-tagatose 3-epimerase PcDTE-IDF8 has shown highly similar reaction mechanisms to that mentioned above [98]. The superimposition of the crystal structure (PDB:4PFH) of this enzyme is shown in Figure 5B.

### 3.3. Mutational Modification Studies of Enzymes

Environmental pollution is a near-inevitable side effect of chemical synthesis. Nowadays, the more favored strategy is to turn d-fructose into a stereoisomer with enzymatic biotransformation routes. This may bring greater social and environmental benefits, but it will also introduce new difficulties because of the inferior biotransformation efficiency, and defects in the thermostability of these enzymes may hamper their industrial perspective [99].

Thus, the prerequisite for a high conversion ratio is to improve the enzymatic activity and thermostability of ketose 3-epimerases. Researchers have implemented protein engineering strategies, with the help of intensive studies on enzymatic crystal structures and active sites, to improve the enzymatic activities and thermostability, as well as for the extension of their half-life.

#### 3.3.1. Site-Directed Mutagenesis

Site-directed mutagenesis and high-throughput screening facilities are among the common methods for obtaining superior mutants of enzymes. Based on the reported three-dimensional structures of ketose 3-epimerases, researchers combined structural modeling and amino acid sequence comparison to select mutation sites and substitute amino acid residues in the target enzymes. The selected multiple amino acid residues for targeted mutations were fulfilled by PCR techniques for the introduction of mutations and for the generation of mutant strains resistant to high temperatures. Kim et al. used error-prone PCR to construct a double mutant (I33L-S213C) of DPEase (from *A. tumefaciens*) with a 29.9-fold increased half-life at 50 °C [96]. Zhu et al. [100] characterized the d-tagatose 3-epimerase from *Sinorhizobium* sp. and analyzed its crystal structure, elucidating that Glu154, Asp187, Gln213, and Glu248 formed a hydrogen-bonding network with the active-site for Mn^2+^ and constituted a catalytic tetramer, while Arg65 and Met9 interacted with the d-fructose and d-tagatose, in which the O-4 positions of Arg65 and Met9 form a unique interaction with d-fructose and d-tagatose. Zhang et al. [24] selected eight amino acid residues for targeted mutagenesis, which were found at the interface region of *Dorea* sp. DAEase by homology modeling analysis, including Phe154, Asn187, Glu191, Ile193, Cys212, Met248, Asp255, and His257. In this report, three mutants (F154Y/E191D/I193F) were screened out and these mutants displayed a higher thermal stability whose t (1/2) values at 50 °C increased by 5.4-fold compared with the wild-type enzyme [24]. Bosshart et al. employed the directed divergent evolution method and obtained the thermostable variant (Var8) of *P. cichorii* DTEase, and exploited it as an efficient catalyst for the C-3 epimerization of d-fructose to d-allulose, and L-sorbose to L-tagatose [98]. In this report, the strategy of iterative randomization and screening around the substrate-binding site was used, and the eight-sites mutant IDF8 had a nine-fold improved *k_cat_* for the epimerization of d-fructose, while the six-sites mutant ILS6 had a 14-fold improved epimerization efficiency for L-sorbose.

Using mutant strains for bioconversion can also significantly shorten the reaction time in whole-cell processes. For instance, Park et al. used a recombinant *E. coli* strain with a double-mutant DPEase and achieved a conversion capacity of 700.0 g/L d-fructose to 230.0 g/L d-allulose in only 40 min under reaction conditions at 60 °C and pH 8.5 [101]. Using systematically optimized interfacial interactions for the promotion of the thermal stability of polycrystals is also an effective protein engineering strategy [102]. The strong interactions resulted in tightly bounded tetramers, which contributed to the thermostability of the ketose 3-epimerase for optimal catalytic temperatures [103].

#### 3.3.2. B-Factor Iterative Testing

The B-factor iterative testing strategy (B-FIT; also known as the temperature factor or the atomic displacement parameter or the Debye−Waller factor) is used in protein crystallography to describe the attenuation of X-ray or neutron scattering due to thermal motion [104,105]. Owing to the intercrystalline thermal motion of proteins, B-factor can be acquired from X-ray data or neutron scattering data, which are often used for the analysis of the flexibility or stiffness of atoms, side chains, or even whole regions [105]. It has been reported that residues located around the binding pocket usually exhibit higher B-factors or disordered regions, and mutations from flexible to rigid may be an effective way to improve the thermostability of the enzyme [106]. B-factor analysis has now been widely developed and applied to protein engineering [107]. The B-factor is often used as an efficient method for genetic engineering, with help of the programs such as Rosetta Design [108], in order to enhance the kinetic and thermodynamic stability of proteins, based on rational design or structure-guided directed evolution and proline substitutions [109]. These methods combine the B-factor analysis with computer calculations, which greatly accelerates the high-throughput screening process.

The most prominent development is the introduction of B-factor iterative testing (B-FIT directed evolution method) using mutagenesis to improve the thermal stability of enzymes in industrial applications [110]. Mao et al. used a d-allulose 3-epimerase from *Pseudomonas aeruginosa* as a template for homologous structure-based targeted mutagenesis, replacing residues in the highly flexible region based on B-factor analysis, with which the conversion ratio was increased to 28.6% [31]. These results suggested that the fixed-point mutagenesis based on B-FIT would be an efficient strategy to improve the thermal stability of ketose 3-epimerase.

### 3.4. Other Enzymes

Besides the d-allulose 3-epimerase, other types of enzymes can also achieve similar functions. L-rhamnose isomerase (LRI, EC 5.3.1.14) is a type of aldose−ketose epimerase that catalyzes the isomerization of d-allose and d-allulose [111], but it exhibits a comparatively low catalytic activity to d-allulose. The crystal structure of LRI from *Methylomonus* sp. has been analyzed and can subsequently be used to obtain mutants [112].

As a complement to the Izumoring strategy, the aldol addition reaction, which relies on the aldolase catalysis of DHAP, provides an efficient method for the synthesis of rare sugars. Some scholars have attempted to generate the intermediate product DHAP through the oxidation of the substrates using cascade catalysis with phosphoglycerol oxidase (GPO) and glycerol kinase (GK). Because of the high price of DHAP, these reactions from cheap raw materials can reduce the cost of feedstock. Li et al. used l-rhamnulose-1-phosphate aldolase from *Thermus thermophilus* HB8 and *Thermotoga maritima* MSB8 for the production of four rare sugars, including d-allulose, using a one-pot four-enzyme system [113,114]. In such reactions, l-glycerophosphate (L-GP) was oxidized by GPO to produce DHAP, and the resulting DHAP was then coupled with glyceraldehyde using L-fuculose-1-phosphate aldolase (FucA) to generate d-allulose-1-phosphate, for which the product then proceeded to remove the phosphate group under acidic conditions by acid phosphatase for the production of d-allulose [113,114].

### 3.5. Enzyme Expression Systems

#### 3.5.1. Enzyme Expression in Prokaryotic Systems

As d-allulose is very rare and is not easily extracted from natural materials, the construction of microbial expression systems is of significant value in enzymatic biotransformation production and for the physiological application of d-allulose. Ketose 3-epimerases from different origins have been successfully expressed in both prokaryotic and eukaryotic expression systems (Table 3), while heterologously expressed epimerases have enzymatic properties more suitable for industrial production.

Prokaryotic expression systems most commonly use *Escherichia coli* and *Bacillus subtilis*. The *E. coli* expression system has the fastest expression speed and a clear background that is easy for genetic manipulation [115]. However, this system usually results in low expression levels or premature translation terminations if the heterologous protein has a large number of consecutive rare codons, so that the target protein forms inclusion bodies or aggregates in the cytoplasmic space that require complicated refolding processes [116]. Fortunately, ketose 3-epimerase expressed in *E. coli* usually do not form inclusion bodies, which results in a high-efficiency soluble overexpression. The recombinant *A. tumefaciens* DPEase was successfully expressed in *E. coli* BL21(DE3), and 230 g of d-allulose was produced from 700 g of d-fructose at pH 8.0 and 50 °C for 100 min [19]. *B. subtilis* is one of the most used hosts in industrial enzyme production at present, and it displays no codon preferences and does not produce inclusion bodies [117]. Su et al. used DPEase derived from *C. cellulolyticum* H10 and inserted its coding gene into the vector pHY300PLK for the construction of a food-grade *B. subtilis* expression plasmid [118]. Zhang et al. first utilized the whole-cell transformation process for the production of d-allulose by *B. subtilis* containing an engineered DTEase from *R. baltica* SH1 [85]. The DPEase from *Ruminococcus* sp. was also expressed in the food-grade microorganism *Bacillus pumilus* and was successfully secreted into the supernatant, which decreased the production cost and facilitated the process of enzyme purification [89].

#### 3.5.2. Enzyme Expression in Eukaryotic Systems

There are few reports on the successful heterologous expression of d-allulose 3-epimerases in eukaryotes. It is speculated that DAEase genes are mainly derived from prokaryotes such as bacteria, whose protein expression, processing, and modification pathways are different from those in eukaryotes. Due to the obvious differences in protein modification and folding processes between the microorganisms in the two domains, misfolding and excessive glycosylation may lead to changes in the molecular weight of the target protein and even to inactivation of the enzyme. Nevertheless, it is worth considering the industrial production of d-allulose with DPEase engineered in yeast for its food safety, enzyme activity, and production cost. Juneja et al. successfully expressed DPEase in *Saccharomyces cerevisiae* KAM-2, which converted d-fructose to d-allulose at 55 °C with a conversion efficiency of 26.6% [119]. *Kluvyeromyces marxianus* has shown a superior thermal stability compared with *S. cerevisiae,* owing to the higher temperature required for DPEase catalysis [120,121,122]. This type of yeast assimilates a variety of cheap substrates and grows rapidly at 45–52 °C [123]. Yang et al. transformed the DPEase gene from *A. tumefaciens* into the heat-resistant *K. marxianus* and obtained a conversion capacity of 190 g/L d-allulose with 750 g/L d-fructose as the substrate, at 55 °C in 12 h [92].

### 3.6. Double Enzyme Coupling and Construction of the Gene Expression Cassette

To explore more industrial production approaches of d-allulose, researchers have attempted to investigate enzymatic processes for the simultaneous production of d-allulose and other products, by coupling the expression of DAEase with other enzymes. d-glucose isomerase (GI) and d-allulose 3-epimerase are employed to convert glucose and fructose into d-allulose via a multi-enzyme cascade [124]. Bosshart et al. [125] used two engineered DTEases for the efficient production of d-allulose and L-tagatose. Besides dual enzyme coupling, some scholars have also attempted to construct multi-gene expression cassettes to improve transformation efficiency. He et al. constructed a tandem expression system for expressing DPEase with repetitive target genes [126]. The recombinant integrative plasmid pDG-nDPE (*n* = 1, 2, and 3) containing one, two, or three consecutive P43-DPEase tandem repeats, respectively, was integrated into the genome of *B. subtilis*, which increased the number of target genes for improved expression levels. Yang et al. selected DAEase genes from *Paenibacillus*
*senegalensis*, *C. cellulolyticum*, and *Ruminococcus* sp. to construct 17 different expression cassettes in *Corynebacterium glutamicum* [127]. Among all of these recombinant strains, DAE16 containing three DAEase genes in the expression vector showed the highest specific enzymatic activity of 22.7 U/mg. L-rhamnulose kinase also has the activity of phosphorylating ketoses. Coupling it with DAEase can break the reversible balance of the isomerization reaction and improve the yield of d-allulose [128].

### 3.7. Biotransformation of d-allulose Based on DAEase/DTEase

#### 3.7.1. Biotransformation Using Immobilized Enzymes

The immobilization of enzymes is an important approach for improving the thermal stability of enzymes. Compared with free enzymes, immobilized enzymes offer the advantages of greater tolerance to environmental changes, easier product separation, increased application frequency, and rapid termination of the enzymatic reaction [129]. In general, the carriers for enzyme immobilization include calcium alginate, chitosan spheres, and various resins. Bioreactors are used to load the immobilized enzymes into columns for the continuous production of d-allulose. Akihide et al. immobilized DAEase on a type of ion exchange resin and maintained the enzymatic activity for four months [130]. Samir et al. first used graphene oxide as a carrier for *Agrobacterium rhizogenes* DPEase immobilization, for which the measures extended its half-life at 60 °C from 3.99 min to 720 min, compared with the free enzyme [131]. In this work, the bioconversion efficiency was also significantly improved, and the graphene oxide immobilized DPEase was used for up to 10 cycles. The results of the covalent immobilization of Smt3-d-allulose 3-epimerase onto functionalized iron oxide magnetic nanoparticles revealed that the iron-enzyme nano bio-conjugates had a significant thermal stability, for which the enzymes had a four- to five-fold extended half-life at 50–65 °C [132]. DAEase from *C. scindens* 35704 was immobilized on an amino-epoxy resin scaffold, mentioned as ReliZyme HFA403/M (HFA), and its specific activity reached 103.5 U/g after four steps (ion exchange, covalent binding, glutaraldehyde cross-linking, and glycine blocking) in the immobilization process. After 2 h of incubation at 60 °C, the activity of the immobilized DAEase retained 52.3% of the relative activity, while the free enzyme activity decreased to 12.5% [133]. Due to the cytotoxic nature of the cross-linking agent glutaraldehyde, safer cross-linking agents are needed for the production of immobilized d-allulose. Furthermore, functionalized polyhydroxyalkanoate nano-beads were used for the immobilization of enzymes, for which the products had a high bioactivity in complex reaction systems [134].

As for the immobilization of the enzymes on microorganisms and their biotransformation, several researchers obtained positive results. He et al. immobilized DPEase on the surface of *B. subtilis* spores and fused DPEase to the C-terminus of the anchoring protein CotZ using peptide junctions [135]. *S. cerevisiae* spores can also be used as enzyme immobilization vehicles. For example, the unique structure of the *S. cerevisiae* spores’ dityrosine layer in the spore wall can accommodate soluble enzymes. The activities of the enzymes encapsulated by the dimeric layer can be enhanced in the osw2Δ mutant, owing to the minor defects of the dityrosine layer [136]. Li et al. used biological and chemical methods to immobilize d-xylose isomerase and DPEase on *S. cerevisiae* spores [137]. The immobilized enzymes showed obvious improvements in heat resistance, storage stability, and recirculation efficiency.

Thus, immobilization strategies can be considered to increase the yields and reduce the corresponding costs for the industrial production of d-allulose. However, the relevant research is still at a rudimentary state, with a relatively low recovery of enzyme activity and complex methods, which can be improved by exploring more suitable materials and accessible methods.

#### 3.7.2. Biotransformation Using Whole-Cells

Biotransformation using immobilized DPEase involves complex processes such as enzyme production, purification, immobilization, and recycling, which can lead to high costs and a low efficiency [87]. In contrast, whole-cell reaction systems have commercial viability and can provide resistance to environmental disturbances, enhance enzyme stability, and avoid multi-step operations [93]. Microbial whole-cells also provide cytoplasmic coenzymes that assist in the main processes and facilitate enzymatic reactions. d-Allulose can be produced by whole-cell reactions, as monosaccharides can freely pass through the cell membrane. However, there are also problems with using whole-cell catalysis reactions. For instance, if the reaction conditions are severe, the cells used for the reaction are easily destructed. Wang et al. immobilized *B. subtilis* whole-cells that expressed recombinant DPEase onto a gel bead biocatalyst using Ca-alginate. The immobilized biocatalyst showed a higher thermal/pH stability and storability with a d-allulose productivity of 32.83 ± 2.56 g/L, while the free cells produced only about 10.44 ± 0.07 g/L [138].

Whether immobilized enzymes or whole-cells are used for the production of d-allulose, the crude products are mixtures containing d-fructose and other chemicals. To obtain rare sugars of commercial-grade purity, the product must be desalinated and decolorized, making the purification process extremely expensive. Li et al. constructed a reaction purification system consisting of two continuously stirred tank reactors, which respectively contained immobilized glucose isomerase and glucose oxidase for the removal of d-fructose. In this system, d-fructose is converted to gluconic acid and is easily separated from d-allulose by anion exchange resins [139].

## 4. Production of d-allulose

The integration of biological processes using cheap raw materials and their cost-effective conversion into high-value biomolecules is essential for the development of bioresource technologies and environmental protection. In recent years, researchers have been working on the conversion of d-allulose with cheap raw materials that contain a lot of sucrose. Inulin has been widely applied in the production of various valuable products, like high-fructose syrup and fructooligosaccharides, through enzymatic or microbial fermentation. Li et al. developed a one-pot, two-enzyme system for the production of d-allulose from inulin, for which the system included *A. piperis* exo-inulinase and *Dorea* sp. DAEase [140]. Under optimized conditions (pH 6.0, 60 °C), a massive production of d-allulose was achieved with a productivity of 21.4 g/L from 100 g/L inulin. Zhu et al. [141] cascaded the reaction in one pot with an exo-inulinase from *Bacillus*
*velezensis* and a DAEase from *Ruminococcus* sp. to produce high-fructose syrup containing d-allulose from inulin in Jerusalem artichoke tubers.

The conversion ratio with cheap renewable agricultural residues, including non-grain crops as the raw material, for the industrial production of d-allulose, has been reported to be generally in the range of 25–30% (*w*/*w*) [85]. As carbon and nitrogen sources are critical components for the synthesis of microbial cell proteins and nucleic acids [142], it is possible to alleviate the carbon catabolite repression effect (CCR effect) during high-density fermentation. In addition, the fermentation process can be optimized by modification of promoters or by knockout of the relevant blocking effect genes in the host bacteria. The combination process of low carbon source concentration and constant feed rate was established in a 3.6 L fermenter to reduce the CCR effect during the fermentation of recombinant *B. subtilis*, resulting in a DPEase activity level (2246 U/mL) that was 15 times higher than that obtained in shake-flask fermentations (148.9 U/mL) [118].

## 5. Conclusions

With the booming of the global economy, the dietary structure of human beings has been changed profoundly. Excessive dietary energy intake leads to obesity and other diseases, which has encouraged the finding and application of low-calorie sweetener substitutes worldwide. d-allulose has gained increasing attention due to its nearly zero-calories property and multiple physiological functions. Enzymatic synthesis is the dominant strategy for the green industrial production of d-allulose, and novel enzymes for ketose 3-epimerization have been continuously screened and studied in recent years. The increasing demands for rare sugars as healthy food additives in the global market have prompted urgent breakthroughs in terms of enzyme property modifications and preparation process optimizations, thus requiring an in-depth understanding of the enzymatic characteristics. The majority of current research is based on the production of rare sugars from fructose as a substrate; however, compared with d-fructose, it is cheaper to produce d-allulose with fructose syrup and fruit residues as substrates, which is conducive to the industrial production of d-allulose in the future. The industrialization and commercialization of d-allulose was first achieved in Japan and Korea [143]. d-Allulose shows great development potentials, while the related research for industrialization is still very weak in other countries and regions.

There are plenty of problems that still exist in the application of current production technology, such as the excessive redundant research works that mainly focus on the upstream gene cloning, recombinant strain construction, and expression of the related enzymes, while the downstream technologies for separation, purification, and commercial production are still insufficient. Therefore, we need to develop easier and more effective methods to produce d-allulose that can facilitate the subsequent recovery and reduce the operational costs of crystallization or recrystallization, isolation, and purification. Fortunately, the advances of synthetic and systems biology and their related technologies will provide more possibilities for the future commercial production of d-allulose. Such advances and developments will greatly benefit future applications of d-allulose in a wider range of food products and through more health interventions.

## Figures and Tables

**Figure 1 foods-10-02186-f001:**
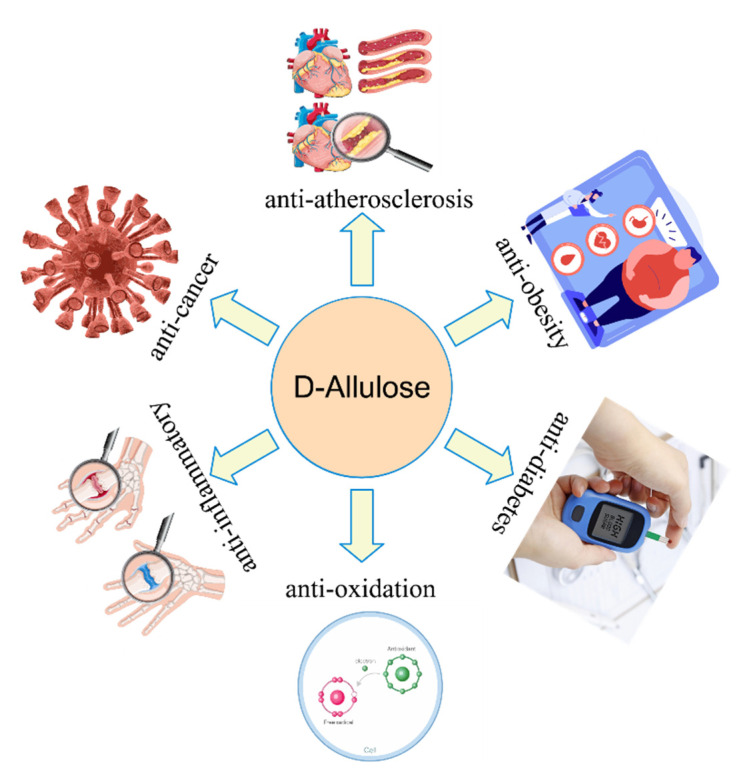
Multiple physiological functions of d-allulose.

**Figure 2 foods-10-02186-f002:**
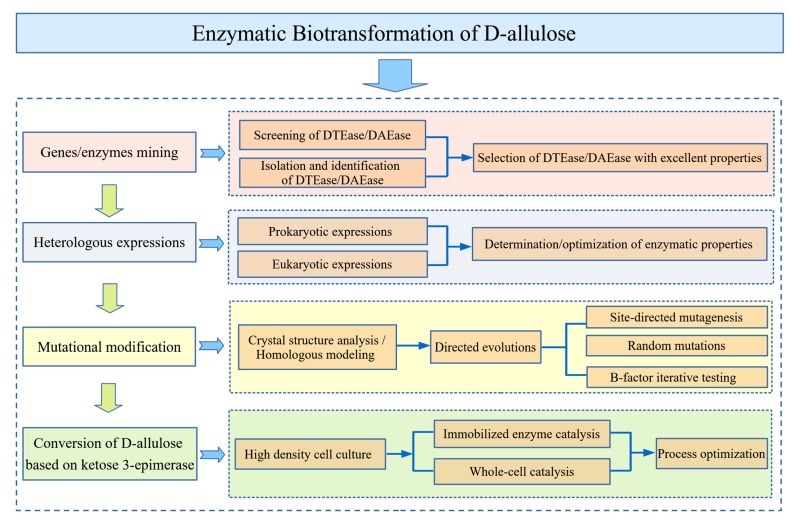
Overview of enzymatic biotransformation processes of d-allulose based on d-tagatose 3-epimerase (DTEase)/ d-allulose 3-epimerase (DAEase).

**Figure 3 foods-10-02186-f003:**
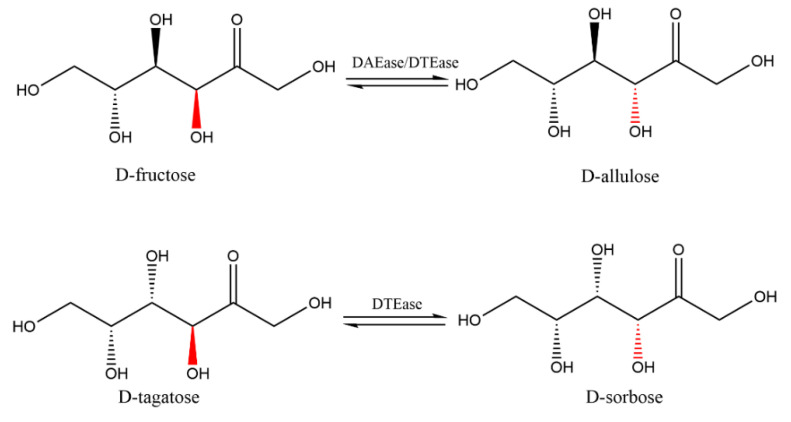
The epimerization reactions of substrates d-fructose and d-tagatose catalyzed by the DAEase/DTEase family.

**Figure 4 foods-10-02186-f004:**
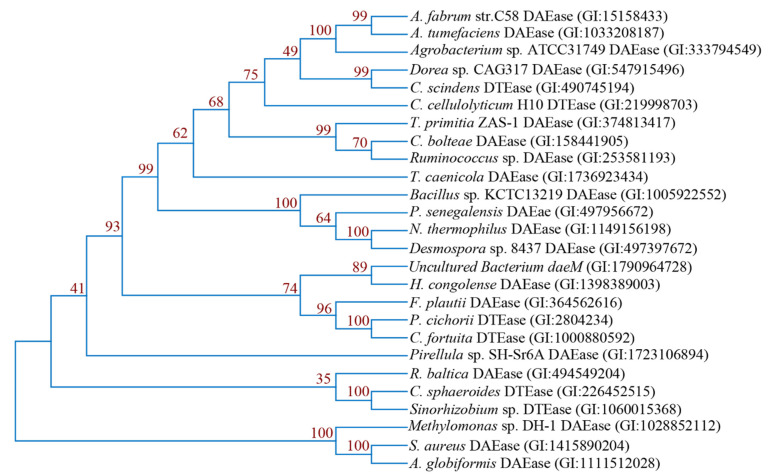
Phylogenetic tree of d-allulose-3 epimerases generated based on the amino acid sequences, using the MEGA X program with the neighbor-joining method.

**Figure 5 foods-10-02186-f005:**
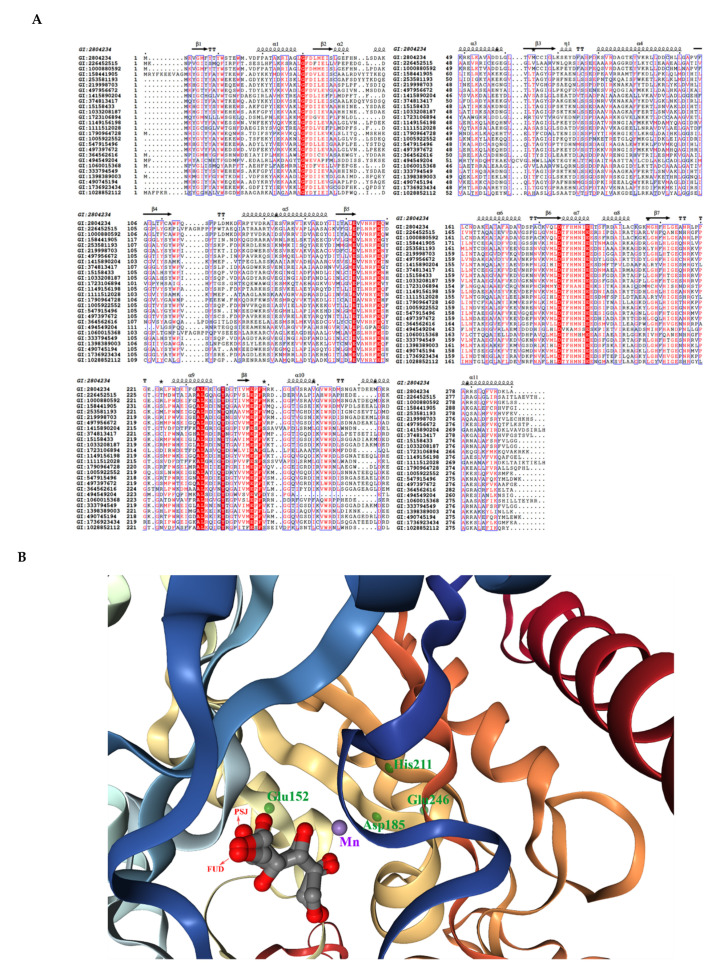
(**A**) Multiple sequence alignment of d-tagatose-3 epimerases and d-allulose-3 epimerases. The red-boxed characters are the strictly identical amino acid residues. (**B**) Superimposition of the crystal structure of PDB:4PFH. The purple ball is the Mn^2+^ ion. The green dots are the ligand residues (in Chain B) with interactions with the Mn^2+^ ion and the substrate. The substrate d-fructose (FUD) and the product d-psicose (PSJ) are shown with arrows. The figure was visualized with an NGL engine powered by the MMTF program.

**Table 1 foods-10-02186-t001:** Detailed chemical properties of d-allulose.

Chemical Properties	Description
Molecular formula	C_6_H_12_O_6_
CAS	551-68-8
Molar mass	180.156 g/mol
PubChem CID	90,008
Physical state	White solid crystal
Crystallize	β-D-pyranose
Conformation	1C (^1^C4 (d))
Smell	/
Melting point	96 °C
Optical rotation	ca. −85 degdm^−1^∙g^−1^∙cm^3^
Solubility	Dissolve 291 g in 100 g water
Sweetness (relative to sucrose)	70%
Energy	0.007 kcal/g

**Table 2 foods-10-02186-t002:** The possible mechanisms of d-allulose in various physiological functions.

Physiological Functions	Possible Mechanisms
Anti-obesity	Inhibition of the synthase activity for fatty acid production; increase of energy expenditure [45], and inhibition of 3T3-L1 preadipocyte differentiation and lipid accumulation [47].
Anti-diabetes	Inhibition of intestinal α-glucosidase [53], suppression of the glycemic response upon carbohydrate ingestion [53], and reduction of postprandial glucose levels [56,57].
Anti-atherosclerosis	Inhibition of the expression of monocyte chemotactic protein-1 [43], decrease of serum PCSK9 levels [49], and enhancement of hepatic HDL-cholesterol uptake via SR-B1 [58].
Anti-inflammatory	Decrease of the expression levels of gene Gm12250 [44].
Anti-oxidation	Effective scavenging of reactive oxygen radicals [59].

**Table 3 foods-10-02186-t003:** Microbial sources and the related information for DTEases and DAEases.

	Sources of Enzyme	GenBank	Expression Host	Expression Cassette	Molecular Weight of Native Protein	Optimal pH	Optimal Temperature (°C)	Metal Ions Required	Conversion Ratio	Ref
DTEase	*P.cichorii* ST24	BAA24429	*E. coli*.JM105	pIK-01-DTE	33	7.5–8.0	60	Mn^2+^	20%	[84]
	*R.sphaeroides* SK011	ACO59490	*E. coli* BL21(DE3)	pET-22b (+)-DTE	128 (tetramer)	9.0	40	Mn^2+^	23%	[23]
	*C.scindens*	WP_004607502.1	*E. coli* BL21(DE3)	pET-22b (+)-DTE	31	7.5–8.0	60	Mn^2+^	28%	[20]
	*C. fortuita*	WP_061137998.1	*E. coli* BL21(DE3)	pET-22b (+)-DTE	70 (dimer)	7.5	65	Co^2+^	29.4%	[78]
	*R. baltica* SH 1	NC_005027	*B. subtilis* WB800	pP43NMK-DTE	32	7	35	NR	25.86%	[85]
	*C. minuta*	NZ_CP029256.1	*E. coli* Rosetta (DE3)	pANY1-DTE	33	6	50	Ni^2+^	30%	[25]
	*Sinorhizobium* sp	NZ_CP016452.1	*E. coli* BL21(DE3)	pET-28a (+)-DTE	68 (dimer)	8.0	50	Mn^2+^	34%	[86]
DAEase	*Clostridium bolteae*	CLOBOL_00069	*E. coli* BL21(DE3)	pET-22b (+)-DAE	139 (tetramer)	7.0	55	Co^2+^	28.8%	[87]
	*Dorea* sp. CAG317	FR892665.1	*B.subtilis* WB800	pSTOP1622-P43- DAE	32.7	6.0	70	NR	NR	[88]
	*Ruminococcus* sp.	ZP_04858451	*Bacillus pumilus*	pNCMO2-DAE	128 (tetramer)	8.0	60	Mn^2+^	26%	[89]
	*Treponema primitia* ZAS-1	ZP_09717154.1	*E. coli* BL21(DE3)	pET-22b (+)-DAE	33.3	8.0	70	Co^2+^	28%	[22]
	*Thermoclostridium caenicola*	SHI77623.1	*E. coli* BL21(DE3)	pET-22b (+)-DAE	33	7.5	65	Co^2+^	28%	[90]
	*A.globiformis* M30	BAW27657.1	NR	NR	128 (tetramer)	7.0–8.0	70	Mg^2+^	24%	[91]
	*A. tumefaciens* EHA 105	KX098480.1	*K. marxianus*	pRS42H-DAE	33	8.0	55	Mn^2+^	25.3%	[92]
	*Flavonifractor plautii*	EHM40452.1	*C. glutamicum ATCC 13032*	pEKEx2-DAE	33	7.0	65	Co^2+^	31%	[93]
	*Pirellula* sp. SH-Sr6A	WP_146677337.1	*E. coli* BL21(DE3)	pET-22b (+)-DAE	32	7.5	60	Co^2+^	31.44%	[94]
	*C.cellulolyticum* H10	NC_011898	*E. coli* BL21(DE3)	pET-22b (+)-DAE	33	8.0	55	Co^2+^	32%	[79]
	*Novibacillus thermophilus*	WP_077721022.1	*E. coli* BL21(DE3)	pET28b (+)-DAE	130 (tetramer)	7.0	70	Co^2+^	29.7%	[95]

Note: NR = not reported.

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
