# Peer review of "Research Advances of d-allulose: An Overview of Physiological Functions, Enzymatic Biotransformation Technologies, and Production Processes"

_foods, 2021, doi:10.3390/foods10092186_

Round 1
Reviewer 1 Report
This review is in general terms well written, is comprehensive, complete, clear, and gathers a representative collection of examples previously reported in the literature. The paper has an important educational component that is of high value. The rare sugar D-allulose has in fact a significant role in different fields. There are some minor revisions.
I have some important comments:
- The author should change in the overall manuscript the word: piscose with psicose (i.e. line 43, 308, 624, 625, fig 3.1)
- Correct ketohelose with ketohexulose line 43
- Some sentences are not clear, please check the right verbs and clarify the concepts: line 23; 56-61; 71 (which?); 198-199;
- Check the abbreviations i.e. DTEase (line 243) should be specified before.
- table 1: b-D-pyranose, change with beta....
- line 312 : the potential revolutionary...etc is a strong affirmation, please revise.
Author Response
1. The author should change in the overall manuscript the word: piscose with psicose (i.e. line 43, 308, 624, 625, fig 3.1)
Response: Thanks to the reviewer. We were sorry for our carelessness, and we have carefully checked the text thoroughly and correct the spelling errors. The “piscose” has been corrected as “psicose” on lines 42, 72, 256, 321, 644, 646, and Figure3 in the revised manuscript.
The revised words were highlighted with a yellow background.
2. Correct ketohelose with ketohexulose line 43
Response: Thank you for pointing this out. It has been corrected as“ketohexulose” in line 42 (Page1) and was highlighted with a yellow background.
3. Some sentences are not clear, please check the right verbs and clarify the concepts: line 23; 56-61; 71 (which?); 198-199
Response: We thank the Reviewer for carefully reading and pointing out the mistakes in our work. We changed more clearly related expressions and as follows:
Line23: The sentence has been changed to “However, there are few comprehensive reviews on this topic” (Page1, Line 22).
Line 56-61: The original expression was indeed not clear. The sentence has been changed to “There are mainly two chemical production processes for this compound: one is synthesis from glucose or 1,2:4,5-di-O-isopropylidene-β-D-fructopyranose under high-temperature reaction conditions catalyzed by molybdate [16,17]; the other is the preparation of D-allulose from fructose by heating in a mixed system of alcohol and triethylamine, with multi-step reactions [18].” in the revised manuscript (Page 2, Line 60-64).
Line 71: The incorrect usage of “which” has been changed to “whose” (Page 2, Line 73) in the revised manuscript;
Line 198-199: The sentence has been changed to “The addition of D-allulose can make products having stronger water holding capacity in foods compared that to sucrose”(Page6, Line209-210).
The revised sentences were highlighted with a yellow background.
4. Check the abbreviations i.e. DTEase (line 243) should be specified before.
Response: Thanks to the reviewer. We have carefully checked all abbreviations in the contexts and corrected the errors. D-tagatose 3-epimerase (DTEase) has been specified in line 246 (Page 7), and D-psicose 3-epimerase (DPEase) has been specified in line 73 (Page 2). We also explained the full name of “FucA”, which was “L-fuculose-1-phosphate aldolase” (Page 13, Line 409-410).
The revised words were highlighted with a yellow background.
5. table 1: b-D-pyranose, change with beta
Response: Many thanks to the reviewer. We forgot to change the "b" to the beta form. It has been corrected as “β-D-pyranose” in Table1. We are so sorry to make these mistakes, and we have carefully checked the manuscript thoroughly to make sure that there have no careless mistakes in our manuscript.
The revised word was highlighted with a yellow background.
6. line 312 : the potential revolutionary...etc is a strong affirmation, please revise.
Response: We thank the Reviewer for the helpful suggestion. Based on the reviewer’s suggestion, the sentence has been changed to “Nowadays, the more favored strategy is to turn D-fructose into a stereoisomer with enzymatic biotransformation routes.”
The revised sentence was highlighted with a yellow background (Page12, Line 324-326).
Reviewer 2 Report
This manuscript reports the review on research advances of D-Allulose about an overview of physiological functions, enzymatic biotransformation technologies, and production processes. Overall, the review seems to be well-written to focus on the enzymatic biotransformation technologies, and production processes. However, there are still some suggestions and/or questions to improve the manuscript as below.
- In the introduction, descriptions on pharmacokinetic and metabolic properties of D-Allulose in vitro and in vivo animal and/or human from previous literatures are needed to understand what happens after taking D-alluose in body.
- For sections 2, the reviewer suggests to add a Table to summarize the important points for each pharmacological function of D-allulose for the readers.
- For Figure 2, the letter is better to write to be parallel, not slanted style.
- In the conclusion, it is strongly suggested to add authors’ outlook more extensively.
Author Response
1. In the introduction, descriptions on pharmacokinetic and metabolic properties of D-Allulose in vitro and in vivo animal and/or human from previous literatures are needed to understand what happens after taking D-allulose in body.
Response: Thanks to the reviewer. This is a very good suggestion. We have added information on the in vivo metabolic pathways and properties of animals after ingestion of D-allulose in the introduction, and as follows: “It has been shown that D-allulose was partly absorbed in the small intestine and released into the bloodstream [3]. While a small portion of D-allulose was delivered to the large intestine and eventually found to be partially fermented in the appendix in rats and to a lesser extent in human intestines [4].” (Page 2, Line 49-52).
The revised sentences were highlighted with a yellow background.
2. For sections 2, the reviewer suggests to add a Table to summarize the important points for each pharmacological function of D-allulose for the readers.
Response: We thank the reviewer for this constructive comment. We have added the suggested Table to the revised manuscript. The possible mechanisms of various physiological functions of D-allulose are summarized in Table 2, and as follows. The Table 2 is in three-line table style in the revised manuscript.
Table 2: The possible mechanisms of D-allulose in various physiological functions
|
Physiological functions |
Possible mechanisms |
|
anti-obesity |
inhibition of the synthase activity for fatty acid production; increase of energy expenditure [45]; inhibition of 3T3-L1 preadipocyte differentiation and lipid accumulation [47]. |
|
anti-diabetes |
inhibition of intestinal α-glucosidase [53]; suppression of the glycemic response upon carbohydrate ingestion [53]; reduction of postprandial glucose levels [57,58]. |
|
anti-atherosclerosis |
inhibition of the expression of monocyte chemotactic protein-1[43]; decrease of serum PCSK9 levels [49]; enhancement of hepatic HDL-cholesterol uptake via SR-B1 [59]. |
|
anti-inflammatory |
decrease of the expression levels of gene Gm12250 [44]. |
|
anti-oxidation |
effective scavenging of reactive oxygen radicals [60]. |
3. For Figure 2, the letter is better to write to be parallel, not slanted style.
Response: Thanks to the reviewer. We carefully checked Figure 2 and revised the annotation. Sorry that we didn't quite understand the specific meaning of “parallel” and “slanted style”. Please tell us where should be modified in the figure, if it is not proper.
4. In the conclusion, it is strongly suggested to add authors’ outlook more extensively.
Response: Many thanks to the reviewer. We think this is an excellent suggestion. Combined with the context, we added some conclusions and prospects about the production of D-allulose in the future. The details added are as following: “The majority of current researches are based on the production of rare sugars from fructose as substrate, however, compared with D-fructose, it is cheaper to produce D-allulose with fructose syrup and fruit residues as substrates, which is conducive to the industrial production of D-allulose in the future.” (Page16-17, Line580-583), and “Therefore, we need to develop easier and more effective methods to produce D-allulose that can facilitate the subsequent recovery and reduce the operational costs of crystallization or recrystallization, isolation, and purification.” (Page 17, Line 591-593).
The revised sentences were highlighted with a yellow background.
Round 2
Reviewer 2 Report
The revised version of manuscript tries to reflect the reviewer's points and/or concern in a proper way.